



# Modelling groundwater recharge, actual evaporation and transpiration in semi-arid sites of the Lake Chad Basin: The role of soil and vegetation on groundwater recharge

Christoph Neukum[1], Angela Gabriela Morales Santos[2], Melanie Ronelngar[3], Sara Ines Vassolo[1]

[1] Federal Institute for Geosciences and Natural Resources, Hannover, 30655, Germany

[2] Institute for Soil Physics and Rural Water Management (SoPhy), University of Natural Resources and Life Sciences, Vienna, 1180, Austria

[3] Federal Institute for Geosciences and Natural Resources, Ndjamena, Chad

*Correspondence to*: Christoph Neukum (christoph.neukum@bgr.de)

**Abstract.**

The Lake Chad Basin, located in the center of North Africa, is characterized by strong climate seasonality with a pronounced short annual precipitation period and high potential evapotranspiration. Groundwater is an essential source for drinking water supply as well as for agriculture and groundwater related ecosystems. Thus, assessment of groundwater recharge is very important although difficult, because of the strong effects of evaporation and transpiration as well as limited available data.

A simple, generalized approach, which requires only a small number of field data, freely available remote sensing data, and well-established concepts and models, is tested for assessing groundwater recharge in the southern part of the basin. This work uses the FAO-dual $K_c$ concept to estimate E and T coefficients at six locations that differ in soil texture, climate, and vegetation conditions. Measured values of soil water content and chloride concentrations along vertical soil profiles at these locations together with different scenarios for E and T partitioning and a Bayesian calibration approach are used to numerically simulate water flow and chloride transport. Average potential groundwater recharges and the associated model uncertainty at the six locations are assessed for the time-period 2003-2016.

Model results show that interannual variability of groundwater recharge is generally higher than the uncertainty of the modelled groundwater recharge. Furthermore, the soil moisture dynamics at all locations are limited by water availability for evaporation in the uppermost part of the soil and by water uptake in the root zone rather than by the reference evapotranspiration.

## 1 Introduction

The Lake Chad Basin (LCB) is one of the largest endorheic basins of the world with an area of approx. 2.5 million km². The basin covers parts of Algeria, Cameroon, Central African Republic, Chad, Libya, Niger, Nigeria and Sudan. According to the Lake Chad Basin Commission (LCBC, 2012), 45 million inhabitants are settled in the basin. The study areas Salamat and Waza Logone are located in the southern part of the LCB, along the major tributary river to the Lake Chad (Figure 1), the Chari-Logone river system, which accounts for around 80-90% of the Lake Chad inflow (Bouchez et al., 2016).





Groundwater is an important source for drinking water supply as well as for agriculture and groundwater related ecosystems
in the LCB. The Lake Chad, the rivers and the floodplains of the major rivers are characterized by strong seasonality, due to a
pronounced short annual precipitation period and high potential evapotranspiration. Groundwater recharge, evaporation,
transpiration, and the entire hydrological budget depend strongly on seasonality. However, the impact of transpiration as a
potentially significant process of the hydrological budget (Jasechko et al., 2013,) has so far rarely been taken into account
(Bouchez et al., 2016).
Many hydrological studies were published concerning the hydrological behaviour and budget of the Lake Chad, due to its
substantial and frequent open water surface changes and related consequences to the population and the environment (e.g.
Bouchez et al., 2016; Lemoalle et al., 2012; Olivry et al., 1996; Vuillaume, 1981). Another important topic associated to Lake
Chad is groundwater recharge by infiltration of lake water into the Quaternary aquifer, which was estimated by isotopes studies
(Fontes et al., 1969; Fontes et al., 1970; Gaultier, 2004; Zairi, 2008), by water and salt budgets (Bader et al., 2011; Carmouze,
1972; Roche, 1980) or hydrogeological models (Isihoro et al., 1996; Leblanc, 2002). Local scale studies focusing on the
hydrological processes in the vadose zone are largely missing in the LCB. Recently Tewolde et al. (2019) published a local
scale study using stable isotope and chloride concentrations in partly the same soil profiles used in this work.
For vadose zone studies, partitioning evapotranspiration (ET) into its respective soil evaporation (E) and plant transpiration
(T) components is crucial for process-based understanding of fluxes (Anderson et al., 2017). There are a number of
measurement and modelling approaches that can be used to estimate E and T separately. Some of the measurements include
micro-lysimeter, soil heat pulse probes, Bowen ratio, and Eddy covariance to determine E; and sap flow, chambers, and
biomass-transpiration relationship to measure T (Kool et al., 2014). Evapotranspiration partitioning can also be estimated
directly by using stable isotopes to assess the ratio between E and T (Wu et al. 2016). Stable isotopes were used in combination
with Eddy covariance on semi-arid environments as well (Aouade et al., 2016).
Anderson et al. (2017) reviewed some recent methodological developments for partitioning ET. These include
micrometeorological approaches involving the flux variance partitioning of high-frequency Eddy covariance observations
(Scanlon and Sahu, 2008, Palatella et al., 2014) and proxies for photosynthesis and transpiration such as measurements of
isotopic fractionation (Griffis et al., 2010) and carbonyl sulfide uptake (Wohlfahrt et al., 2012). They also discussed the
partitioning of energy balance between canopy and soil using remote sensing (Colaizzi et al., 2012; French et al., 2015).
The Food and Agricultural Organization of the United Nations (FAO) published a model (Allen et al., 1998) that uses an
empirically defined crop coefficient (Kc) in combination with a reference ET (ET$_0$) to calculate crop evapotranspiration (ETc).
There are two approaches for this method: single coefficient and dual crop coefficient. The FAO-dual Kc model is a validated
method for ET partitioning and the most commonly applied (Kool et al. 2014). It has been widely used with good results for
numerous crops under different conditions: e.g. wheat and maize in semi-arid regions (Shahrokhnia and Sepaskhah, 2013),
wheat in humid climate (Vieira et al., 2016), cherry trees in temperate continental monsoon climate (Tong et al., 2016), irrigated
eucalyptus (Alves et al., 2013), and canola in terrestrial climate (Majnooni-Heris et al., 2012).



Quantification of water fluxes in the vadose zone and linking atmospheric water and solute input at the upper boundary of the
soil with water and solute fluxes at different soil depths is frequently implemented using different kind of models. Numerical
models need information on vadose zone properties for accurate parametrization to link fluxes with state variables such as
unsaturated hydraulic conductivity and water retention curve. Estimation of effective soil hydraulic parameters, which are
valid at the modelling scale, might be laborious. Furthermore, parameter estimation might vary significantly depending on the
measurement method (Mertens et al., 2005), when water and solute fluxes dynamics are considered. Hydraulic and transport
parameters obtained from inverse modelling can be ambiguous, if multiple parameters are simultaneously considered and
boundary conditions are not well known. Combining different state variables of water flow and solute transport in one objective
function was found to be a strategy for appropriate parametrization (Groh et al., 2018, Sprenger et al., 2015) and for the
transient simulation of water and solute fluxes. However, large amount of data are necessary to obtain accurate estimates of
state variables, which are rarely available in remote areas of Africa, and measurement of related variables are associated with
a huge effort in such environments. Pedotransfer functions (PTF) bridge available and needed data and are frequently used to
quantify soil parameters (van Looy et al., 2017; Vereecken et al., 2016). PTF strive to provide a balance between data accuracy
and availability (Vereecken et al., 2016). Since PTF usually do not consider soil structure, they result better for homogeneous
soils than for structured soils (Sprenger et al., 2015; Vereecken et al., 2010).
Recharge occurs even in the most arid regions, mainly due to concentration of surface flow and ponding with lateral and
vertical infiltration (Lloyd, 1986). Direct recharge by precipitation is possible in semi-arid regions, but intermittently, owing
to the fluctuations in the periodicity and volume of precipitation that is inherent to such regions (Lloyd, 2009). Scanlon et al.
(2006) synthesized recharge estimates for semiarid and arid regions globally. They found that recharge is sensitive to land use
and cover changes, hence management of such changes are necessary to control recharge. Moreover, they state that average
recharge rates in semi-arid and arid regions range from 0.2 to 35 mm yr$^{-1}$, representing 0.1 to 5 % of long-term average annual
precipitation. Edmunds et al. (2002) estimated direct recharge rates from precipitation in the Manga Grasslands in NE Nigeria
(western LCB) at rates between 16 mm year$^{-1}$ and 30 mm yr$^{-1}$. Using the same method, they appraised the regional direct
recharge for north Nigeria at 43 mm year$^{-1}$, which highlights the importance of infiltration from precipitation to the
groundwater table at regional scale. Recently, Cuthbert et al. (2019) investigated the relationship between precipitation and
recharge in sub-Saharan Africa using multidecadal hydrographs. They found that focused recharge predominates in arid areas
and is mainly controlled by intense rainfall and flooding events. Intense precipitation, even during years of lower annual
precipitation, results in some of the largest years of recharge in dry subtropical locations.
The Chloride Mass Balance (CMB) approach is a widely used technique for estimating groundwater recharge. Edmunds and
Gaye (1994) used interstitial water chloride profiles from the unsaturated zone, in combination with measurements of chemical
parameters from dug wells samples, to calculate groundwater recharge in the Sahel (mean annual rainfall 1970-1990 around
280 mm). A recharge rate of 13 mm year$^{-1}$ over the studied area was obtained. They conclude that it is an inexpensive technique,
which can be applied in many arid and semi-arid areas. Tewolde et al. (2019) applied the CMB on soil profiles of the LCB,
which are partly used in this study. They estimated generally lower annual recharge in Salamat (3 to 19 mm year$^{-1}$) compared





to Waza Logone (50 to 118 mm year$^{-1}$). Among others, one major difficulty of CMB is the choice of a representative chloride
concentration for soils, particularly those with a strong vertical chloride concentration variability.
In general, time series of relevant data for estimating groundwater recharge is scarce in the LCB. A simple, generalized
approach, which requires only a small number of field data, freely available remote sensing data, and well-established concepts
and models is tested for assessing groundwater recharge in the semi-arid part of the LCB. This work uses the FAO-dual $K_c$
concept to estimate E and T coefficients at six locations, which differ in soil texture, climate, and vegetation conditions.
Measured values of soil water content and chloride concentrations along vertical soil profiles at these locations are used
together with different scenarios for E and T partitioning and a Bayesian calibration approach to numerically simulate water
flow and chloride transport. Average potential groundwater recharges and the associated model uncertainty at the six locations
are assessed for the time-period 2003-2016.
**2 Data and methods**
**2.1 Study sites**
The LCB is a Mesozoic basin and a major part of its geology comprises sedimentary formations from the Tertiary and
Quaternary periods (LCBC, 1993). The Quaternary sediments form a continuous layer of fluviatile, lacustrine and aeolian
sands. These medium to fine-grained sands act as an unconfined transboundary aquifer, as do the rest of the aquifers in the
LCB, and are isolated from underlying aquifers by a thick layer of Pliocene clay (Leblanc et al., 2007; Vassolo, 2009). The
Tertiary formation (Continental Terminal) consists of sandstones and argillaceous sands and is a classic example of a confined
aquifer system that becomes artesian in the surroundings of the Lake Chad (Ngatcha et al., 2008). The availability of water
from precipitation as well as the deposition characteristics of the aquifer play an important role in the aquifer recharge of the
upper unconfined sands (Vassolo, 2009).
The study sites (Figure 1, Table 1) are located in the floodplains Waza Logone and Salamat in the southern Sahel subzone. In
the Salamat region, millet and sorghum are grown with trees such as Acacia albida, A. scorpioides and A. sieberana present
along the margins of the floodplains (Bernacsek et al., 1992). In the Waza Logone area, the vegetation is classified according
to the duration of submersion, being the grass savannahs flooded for longer periods of time (Batello et al., 2004).
**2.2 Climate data**
Monthly precipitation and potential evapotranspiration data from 1970 to 2019 for the specific sites in Salamat and Waza
Logone are extracted from the CRUTS 4 database (Harris et al. 2020). The potential evapotranspiration is calculated using the
Penman-Monteith method and is considered herein as the reference evapotranspiration ($ET_0$). The wind speed values at 10 m
above ground for Salamat and Waza Logone were obtained from Didane et al. (2017). To adjust these values for 2 m above
ground, a correction factor of 0.7479 was applied, based on a logarithmic wind speed profile (Allen et al., 1998).





Average annual precipitation in Salamat and Waza Logone are 807 mm and 709 mm, respectively. The rainy season is typically
from May to September with maximum precipitations in July and August. Overall, Waza Logone presents higher $ET_0$ (169
mm month$^{-1}$) than Salamat (144 mm month$^{-1}$). Average annual values of $ET_0$ are 1718 mm in Salamat and 2011 mm in Waza
Logone, exceeding annual precipitation by more than twice. However, during the peak of the rainy season (July to September)
the monthly water balance is positive. The average water balance for July until September between 2003 and 2016 is 131 ±
101 mm month$^{-1}$ and 90 ± 63 mm month$^{-1}$ for Salamat and Waza Logone, respectively (Figure 2).
Chloride concentration of precipitation were measured in 42 samples collected in N'Djamena and Waza-Logone areas between
2014 and 2020 for different precipitation events and stages of the rainy season. The precipitation was sampled using a
Hellmann rainwater collector in N'Djamena. Average chloride concentration in May is 2.5 ± 2.3 mg/l (3 samples). Precipitation
in June to September have suggestively lower chloride concentrations declining from 0.6 ± 0.3 mg/l to 0.26 ± 0.12 mg /l and
0.38 ± 0.14 mg/l at the end of the season. Strong rain events in July and August have chloride concentrations between 0.2 and
0.3 mg/l. The annual wet Chloride deposition sums to 1.8 ± 0.2 kg/ha. Dry deposition of chloride is estimated between 10 –
30% of wet deposition (Bouchez et al. 2019). The measured values are in the range of published data (Goni et al. 2001, Gebru
and Tesfahunegn, 2019). However, not all rain samples could be analyzed for chloride concentration, due to limited sample
amount. This is particular true for minor events with low precipitation amounts at the very beginning of the rainy season.
**2.3 Soil and vegetation data**
At each study site, vertical soil profiles are drilled using a hand auger. In Salamat, soil profiles were sampled in 2016 and
2019. In Waza Logone soil samples were sampled in 2017 only, due to security reasons in 2019. Each of the soil profiles were
fractionated into 10 cm intervals and filled into headspace glass vials and plastic bags. Each soil fraction was tested for grain
size distribution using sieving and sedimentation standard procedures, resident chloride concentration, and gravimetric water
content. Chloride concentration was analyzed after aqueous extraction from oven dried (105°C for 24 hours) soil samples in
the plastic bags following the standard guideline DIN EN 12457-1 (Tewolde, 2017). The gravimetric water content was
converted into volumetric water content using typical bulk densities for the different soil types and locations (Global Soil Data
Task Group, 2000). The type of vegetation and the annual cycle of crops, length of the flooding period, and vegetation
throughout the dry period were mapped during field work and documented by surveying resident population. In addition,
MODIS vegetation indices data (Didan, 2015) were used to justify the documented annual cycle of phenology (Figure 3).
**2.4 Partitioning of evaporation and transpiration**
The calculation of reference evapotranspiration $ET_0$ as well as crop evapotranspiration ($ET_c$) and its partitioning into potential
evaporation and transpiration is based on the dual crop coefficient (Kc) method (Allen et al., 1998). The approach requires two
different coefficients the basal crop coefficient (Kcb) that describes plant transpiration and the soil water evaporation
coefficient (Ke) that depicts evaporation from the soil surface. Kcb is defined as the ratio of the crop evapotranspiration over
the reference evapotranspiration ($ET_c/ET_0$), when the soil surface is dry and transpiration occurs at a potential rate (i.e., water


does not limit transpiration). Ke describes the evaporation component of $ET_c$. When the topsoil is wet, Ke is maximal, but
diminishes with drying out of topsoil to become zero, if no water remains near the soil surface for evaporation. The so-called
dual Kc is the sum of Kcb and Ke. The parameters required for the estimation of $ET_c$ are the reference evapotranspiration
($ET_0$), the basal crop coefficient (Kcb) and the evaporation coefficient (Ke):
$ET_c = ET_0 * K_c = ET_0 * (K_{cb} + K_e) \,,$                                 (2)
Onsite information on vegetation and phenology, such as month of planting, full emergence of crop, and harvesting was used
to define the monthly variation of vegetation at the study sites. These different vegetation periods were combined with crop
specific Kcb information for sorghum and grass provided in Allen et al. (1998). Following Allen et al. (1998), the coefficients
Kcb–mid and Kcb–end were adjusted to comply with the local semi-arid climate in Salamat and Waza Lagone. Monthly Kcb
values for acacia were estimated based on the work of Do and Rocheteau (2003) and Do et al. (2008).
Site-specific estimated monthly variation of ground cover and flooding periods with ranges of crop coefficient (Kcb), soil
water evaporation coefficient (Ke), and root depth is provided in Table S1.
**2.5 Modelling water flow and solute transport**
**2.5.1 Model concept, setup, and initial conditions**
The chloride profiles measured in soil represent the input history for water and solute budget from past precipitation events,
which can be estimated by transient water flow and solute transport modelling. The model concept assumes that atmospheric
chloride input is restricted to solute in precipitation and that the chloride concentration profile results from solute enrichment
in the soil, due to evaporation and transpiration. A precise parametrization of the unsaturated flow and transport model and a
robust quantification of groundwater recharge are not possible with the available data and hence cannot be the scope of this
study. However, the model results estimates groundwater recharge magnitude and variability based on information regarding
soil texture and vegetation as well as associated results uncertainty. This approach is appropriate for locations with limited
availability of long-term soil water measurements.
The Hydrus-1D software package was used to simulate transient water flow and solute transport in the six variably saturated
soil profiles. Hydrus-1D numerically solves the Richards (1931) equation for variably-saturated water flow and advection-
dispersion equations for heat and solute transport (Šimůnek et. al, 2009). The processes simulated in the six soils were water
flow, solute transport, and root water uptake for a defined period subdivided in monthly steps. The calculations ended at the
soil sampling time (December 2016 and July 2019 for Salamat and June 2017 for Waza Logone). Root growth was considered
in all the profiles except for ST2, in which the roots of the acacia trees were distributed along the whole profile and assumed
invariant over the simulation period. Because the initial conditions of soil moisture and resident chloride concentration are
unknown, arbitrary values were adopted. To account for the different residence times of water and chloride, due to different
degrees of evapotranspiration and unknown initial conditions, the models encompass different calculation time periods: ST1,
ST2, WL1 and WL2 start in 1990 whereas ST3 in 2010 and WL3 in 1970. To adequately estimate the initial conditions and



reduce the computation time for calibration, a burn-in period of 80 years was considered for ST1 and ST2. All profiles were
discretized into 101 nodes and different horizons according to the soils types interpreted from the individual grain size
distributions.

### 2.5.2 Water flow

For calculation of water retention and unsaturated hydraulic conductivity functions, the Mualem-van Genuchten (MVG) model
(van Genuchten, 1980) was applied. The initial parametrization of these functions was realized using pedotransfer functions
implemented in Rosetta (Schaap et al., 2001), which is a dynamically linked library coupled to Hydrus-1D. The input
parameters for each profile were the percentages of sand, silt, clay, and bulk density at several depths. Whenever several
consecutive layers of a profile showed almost the same grain distribution (texture), the layers were lumped in one, parameter
averages were used in the model, and the measured soil moisture profiles were considered indicatively. The tortuosity
parameter l [-] of the MVG was set to 0.5 in accordance to Mualem (1976).
The upper boundary condition was defined as variable atmospheric condition with surface runoff, whereas the lower boundary
condition was set to zero-gradient with free drainage of water for all profiles, except WL3. At WL3 confined groundwater
conditions prevailed below the confining clay layer encountered at 3.9 m depth. Groundwater was hit at 3.9 m depth, but
rapidly rose to 2.6 m below surface. Consequently, a constant head condition was implemented at that depth.

### 2.5.3 Root water uptake and root growth

The sink term $S$ in the Richards equation, defined by Feddes et al. (1978) as the volume of water removed from a unit volume
of soil per unit time due to plant water uptake, was considered in all soil profiles according to the prevailing vegetation (Table
S1). The Feddes' default parameters for grass were used in ST03 and Waza Logone profiles. In ST01, corn parameters were
selected, since sorghum data is not available. Sorghum and corn roots extract water from approximately the same soil depths
and have similar average root density distribution, in comparison with other crops, e.g. soybeans (Righes, 1980). In the case
of acacia in ST2, the adopted parameters correspond to deciduous trees.
For sorghum in ST1, an average root depth of 1 m was adopted for the initial and end seasons, and 2 m for development and
mid seasons. In ST3 and WL2, the vegetation was defined as grass, while in WL1 and WL3 as grasslands with a flooding
period. Rooting depths values used in these sites range from 0.1 to 0.5, depending on the growth stage of grass. Monthly
variations were specified as time variable boundary conditions. The median maximum rooting depth value of annual grass in
water-limited ecosystems is 0.37 m with a 95 % of confidence in an interval of 0.26-0.55 m (Schenk and Jackson, 2002).
For ST2, the root depth of the acacia tree was considered to be constant over the simulation time with maximum root
distribution at 0.5 m and decreasing distribution down to 2 m (Beyer et al., 2016).


### 2.5.4 Solute transport

The chloride concentration in soil water was simulated using an equilibrium advection-dispersion model implemented in Hydrus1-D. Hydrodynamic dispersion was implemented considering dispersivity values of 10 % of the individual layer thickness in the soil model, a molecular diffusion coefficient of $1.3 \times 10^{-9}$ m²s$^{-1}$, and a tortuosity factor as defined by Millington and Quirk (1961). Adopted dispersivity values are with the reported ranges between 0.8 cm and 20 cm (Vanderborght and Vereecken 2007, Stumpp et al, 2009, 2012).

A third type (Cauchy) boundary condition was applied to the upper and a zero-gradient boundary condition to the lower boundary. The transient liquid phase concentration of the infiltrating water follows measured chloride concentration in precipitation sampled in N'Djamena. Chloride concentration of ponding water in Salamat ranges between 2.5 mg/l and 25 mg/l with an average of 9 mg/l (n=4).

### 2.5.5 Crop evapotranspiration scenario definition

Simulated crop evaporation scenarios and their individual descriptions are provided in Table 2. Parameter ranges for minimum, maximum Kcb and Ke factors are listed in Table S1. The average corresponds to the average value calculated from minimum and maximum factors.

### 2.5.6 Bayesian model calibration

Based on the crop evapotranspiration scenarios, the models were calibrated using a Bayesian calibration referenced by the soil moisture and chloride concentration profiles. We implemented the sum of likelihood functions for soil moisture and chloride concentration to calculate the log-likelihood of a simulation given the observations and standard deviations at each calibration step. The posteriori parameter distribution was estimated using the Differential Evolution Markov Chain Monte-Carlo (DEzs) algorithm with three sub-chains (ter Braak and Vrugt, 2008) implemented in the R package BayesianTools (Hartig et al. 2019). The number of iterations was defined individually according to a Gelman-Rubin reduction factor < 1.2. The calibration was implemented with scaling factors ranging from 0.75 to 1.25 for the MVG parameters saturated volumetric water content, alpha, and n individually as well as chloride concentration and transpiration to account for observed variabilities. However, ranges for MVG model parametern were constraint to n > 1.01. Log-transformed saturated hydraulic conductivity for each layer were considered with ranges from -0.5 to 0.5. The scaling factor for transpiration was used as divisor for evaporation simultaneously to remain within the calculated rate of $ET_0$. From all accepted model runs, 100 were randomly selected at each individual location to evaluate average model results and standard deviations.





## 3 Results

### 3.1 Grain size distribution

Soil textures were defined based on grain size distributions of the six profiles (Figure 4) according to the US Department of
Agriculture soil texture triangle. Most of them are fine-grained soils (clay, sandy clay) and fine-grained soils with minor parts
sand and loam. Only soil profile ST03 is dominated by sand and sandy clay loam.

### 3.2 Model parametrization

The calibrated parametrization of the MVG model for each layer of the six sampling locations are plausible in general (Table
3). The posterior distributions of the Bayesian calibration show the sensitive parameters of the model fit. For ST1, these are
the model parameters n, saturated water content, chloride concentration, and the fraction of transpiration in evapotranspiration.
The saturated hydraulic conductivity is less sensitive (Fig. S1). For ST2, the sensitivities of the model parameters are similar
with the saturated hydraulic conductivity of the upper layer being sensitive and the influence of chloride concentration being
less significant (Fig. S2) compared to the other locations. The model fits of the data from site ST3 are generally insensitive.
Only the model parameters alpha, n, and saturated hydraulic conductivity of the upper layer and chloride concentration in
precipitation show tighter posteriori distributions (Fig. S3). For the model for site WL1, the model parameters n of layers 1, 2,
and 3 and as well as the saturated water content of layers 3 and 5, and subordinately of layer 4, are sensitive. The parameters
alpha for layers 1 to 5 and the chloride concentration and fraction of transpiration are hardly sensitive in the range of the prior
distribution (Fig. S4). For the model for WL2, the parameters n of all layers, the saturated hydraulic conductivity of layer 3,
and the saturated water content of layers 2 and 3 are sensitive (Fig. S5). For WL3, the saturated water contents of layer 2 and
as well as the saturated hydraulic conductivity of layers 1 and 2 and the fraction of transpiration in evapotranspiration are
sensitive (Fig. S6).

### 3.3 Soil water content, chloride concentration and groundwater recharge

Measured and simulated water content and chloride concentration profiles for individual scenarios are shown in Fig. 5. The
average root mean squared error (RMSE) of simulated water content for all individual scenarios ranges from 0.02 to 0.06
(Table 4). In general, the models reproduce well the water content and chloride concentrations. However, the misfit of
measured soil water content to simulated water content is considerable for the models ST1 and partly ST2. The models do not
represent the high chloride concentrations in the uppermost part of soil profiles for ST3, WL1, and WL2. The standard
deviations in chloride concentration of the randomly selected model runs are exceptionally high in the lower part of ST2 that
corresponds to the poor sensitivity of the chloride concentration at the upper boundary and the comparably wide range of
measured chloride concentration in ponding water in the Salamat region (2.5 – 25 mg/l).


The interannual variability of modelled groundwater recharge differs considerably among locations (Figure 6, Table 5). In
general, interannual groundwater recharge variability depends on vegetation and soil texture with related water retention
capacity. Vegetation with deep roots on soil with comparably high water retention capacity have a higher interannual
variability, e.g. ST1, ST2 where recharge occurs only in years with high precipitation. Fine textured soils with shallow rooting
vegetation have an intermediate variability (WL1, WL2, and WL3), where years without recharge occur only during drought
periods. The coarser textured soils with grass cover has low interannual recharge variability (ST3) and recharge occurs each
year. Years with high precipitation, e.g. 2006, 2007, and 2008 in Waza Logone as well as 2010 in Salamat, produce strong
groundwater recharge.
The highest recharge was calculated for ST3 in Salamat, where on one hand water balance during the rainy season (July-
September) is higher compared to the Waza Logone region and on the other hand, shallow rooting vegetation on comparably
coarse soil texture with low water retention capacity and higher hydraulic conductivity prevail. The other locations in Salamat
have lower calculated annual recharge, due to deep rooting vegetation and higher soil water retention capacity. The impact of
soil texture on annual groundwater recharge becomes apparent by comparing the three location in Waza Logone with the same
vegetation on soils with different water retention capacities and hydraulic conductivities. In general, groundwater recharge
expressed as fraction of precipitation is below 8 % (Table 5). Only at ST3, a comparably high fraction of app. 12 % is estimated.

Chloride concentration and water budget of the soils over the modelled time-period are rather unstable and differ for the six
locations. At location ST2 with clay loam soil covered by Acacia and grass, accumulation of chloride takes place over several
years, due to the high transpiration related to the effective field capacity (Figure 7). However, in high precipitation periods,
most of the accumulated chloride is leached to groundwater and soil concentration diminishes. It should be noted that at this
site, the measured chloride concentrations cannot be reconstructed if only input via precipitation is considered. The measured
profile can only be plausibly modelled with an additional input via ponding water. Chloride input at the upper boundary is
consequently much higher at ST2 compared to the other locations considered in this study.
At location ST3, the chloride accumulation is much lower compared to the other locations. Chloride budget is controlled by
the fast groundwater recharge response to precipitation, which flushes chloride from the soil towards groundwater annually.
The majority of chloride infiltrated with precipitation remains in the vadose zone over years and is leached towards
groundwater mainly during years with precipitation or water infiltration above threshold values (Figure 7). Chloride
accumulation is highest in profiles with clay soils and high effective field capacity (ST1, WL1, and WL3).
**3.4 Evaporation and transpiration**
The transpiration amount depends on the availability of water in the root zone and the type of vegetation cover. At ST1, annual
transpiration presents two peaks: one related to sorghum and the other to grass (Figure 8). At each location and in every
simulation year, soil water content in the root zone reaches the wilting point defined by the specific parametrization of the root
water uptake model.





The actual evaporation rate depends mainly on the availability of water in the upper soil zone (Table 6). Clay and clay-loam
with relatively high water storativity have larger amounts of evaporated water compared to sand and loam soils. During dry
seasons, the uppermost part of the soils dries up annually, which restricts evaporation strongly.
Actual evapotranspiration is lower than the reference evapotranspiration most of the year. During and shortly after the rainy
season, when sufficient soil water is available, actual evapotranspiration is comparable to or higher than $ET_0$ depending on the
vegetation.

**4 Discussion**

Soil texture information is helpful to limit possible MVG parameter ranges while searching for realistic parameter sets
(Sprenger et al. 2015). However, poor representation of soil moisture dynamic using MVG parameters derived with ROSETTA
are reported (Sprenger et al. 2015) and indications are given that soil structure has to be taken into account (Vereecken et al.
2010), especially for soils where high rock content influences water flow due to inherent heterogeneity (Sprenger et al. 2015).
The soils at the locations considered in this study belong to Quaternary sediments in the Lake Chad basin and heterogeneity,
due to rock fragments is largely absent. Furthermore, soil moisture dynamics over the year is much higher in soils of flooding
planes compared to soils from the more humid regions in the south, where precipitation although large occurs over 4-5 months
and lacks over the rest of the year. It is expected that high soil moisture dynamics, rather homogeneous soils, and the monthly
resolution of climate data result in minor impact of soil structure on MVG parametrization and groundwater recharge as shown
in chapter 3.2. Soil moisture dynamics at all locations considered in this study are limited by water availability for evaporation
in the uppermost part of the soil and by water uptake in the root zone, but not by the reference evapotranspiration. However,
time resolution of precipitation and evapotranspiration data is monthly and the models probably underestimate soil moisture
dynamics.
Calculated chloride concentrations for the soil profiles give indications of appropriate MVG parametrization as well as
evaporation and transpiration partitioning. However, uncertainty of chloride input and its transient variability in particular is
expressed in rather wide and partly bimodal distribution of the scaling factor (sc_Conc) included in the calibration (Figures
S1-S6 in supplement material). On one hand, measured chloride concentration in precipitation are in agreement with other
studies in central Africa (Goni et al. 2001, Gebru and Tesfahunegn, 2019) and its transient behavior within the rainy season is
considered in the applied model. On the other hand, impact of dry deposition is unknown, because of data scarcity and potential
lateral flow of periodical flooding. Furthermore, due to the monthly resolution of the atmospheric boundary condition, extreme
rain events that cause surface runoff cannot be reflected in the model. The variability of chloride concentration in some of the
soil profiles, which cannot be completely reproduced by the model, indicates either a higher variability of chloride input and/or
a larger variability in soil physics.
Bouchez et al. (2019) identified a chloride deficit between deposition and river export in the Chari-Logone river system of
88 % (only 12 % of the deposited chloride is exported via river water). They refer to the chemical memory effect, which can
play an important role in arid regions. Our simulations show the importance of the vadose zone for storage of chloride over





longer periods of time, which explains on one hand the fate of chloride in the basin and confirms the chemical memory effect.
In this context, it must be noted that the thickness of the vadose zone at the locations considered in this studies is between 4
and 21 m, where important amounts of chloride can be potentially stored leading to a strong delay of the chemical signal from
precipitation to groundwater.

## 349    5 Conclusions

The quantitative estimation of groundwater recharge in the LCB is difficult, due to the scarce data availability and the expected
low recharge quantities. Estimation of low recharge amounts in arid and semi-arid areas are usually ambiguous, because the
immanent measurement inaccuracies lead to uncertainties during data processing and modelling. Quantification of water and
solute fluxes in the vadose zone is often implemented using long-term time series of soil moisture, pressure heads, and
concentration data in combination with appropriate models. Monitoring of soil moisture and solute concentration over longer
periods at different depths and sites is difficult in the LCB, due to limited infrastructural prerequisites and challenging
climatically boundary conditions. The presented approach combines soil moisture and chloride concentration quantified along
vertical soil profiles in different locations within the LCB with numerical models and freely accessible data, while considering
data uncertainty. Although modelling results of soil water content and chloride concentration deviate considerably from
measured values for some profiles, their magnitudes agree largely. This is especially important for chloride concentration in
the middle and deeper parts of the profiles, where seasonal effects are mainly averaged. Thus, the estimates of soil water
balance and especially of groundwater recharge as well as the adopted soil physical parameters are plausible.
Groundwater recharge values estimated in this study are different to those published in Tewolde et al. (2019). This is due to
the more extensive availability of chloride concentration data in precipitation available for this study. In addition, Tewolde et
al. (2019) roughly estimated one value of saturated porosity for each profile. Because this parameter is rather sensitive in the
Bayesian calibration, several values along each of the profiles were considered in this study. In contrast to the assessment of
groundwater recharge with the chloride balance method (Tewolde et al. 2019), the method used here allows not only estimates
of mean recharge, but also its interannual dynamics, variability, and the classification of the uncertainties of the input data and
modelling. The interannual variability of groundwater recharge is generally higher than the uncertainty of the modelled
groundwater recharge. The soil moisture dynamics at all locations considered in this study are limited by water availability for
evaporation in the uppermost part of the soil and by water uptake in the root zone and not by the reference evapotranspiration.
Upscaling of the results to larger areas must be interpreted with caution since the considered combinations of soils and
vegetation probably do not cover all combinations present in Salamat and Waza Logone regions.

## 373    Author contribution

M.R. conducted fieldwork; A.G.M.S. and C.N. conducted modelling and interpretation; C.N. and S.V. design the study and
conducted writing. All authors contributed to the discussion of results and commented the manuscript.





**Acknowledgement**
This study was conducted within the framework of the technical cooperation project "Lake Chad Basin - Management of
Groundwater Resources" jointly executed by the Lake Chad Basin Commission (LCBC) and the German Federal Institute for
Geosciences and Natural Resources (BGR). The technical project is funded by the German Federal Ministry for Economic
Cooperation and Development (BMZ).

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

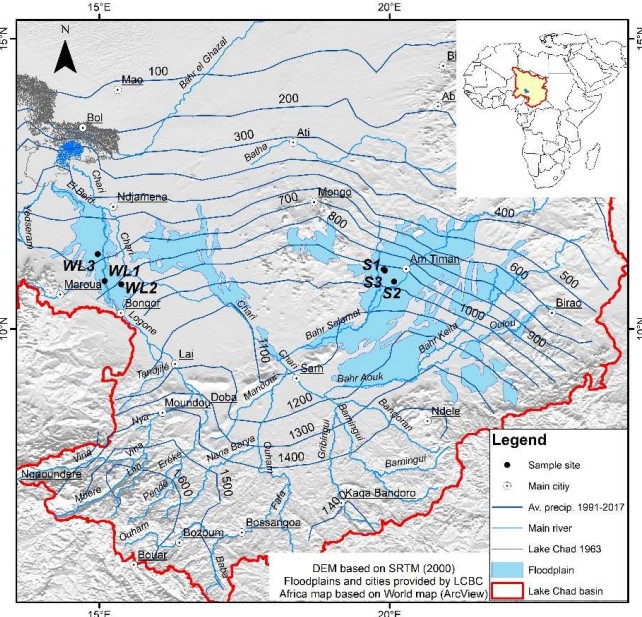


**547 Figure 1: Location of the six soil sampling sites within the Logone and Salamat river basins in the Lake Chad catchment. The map**
**548 inlet shows the location of the Lake Chad basin in Africa.**





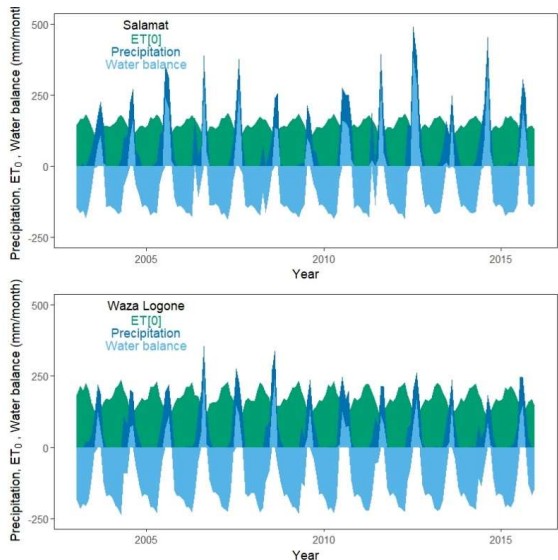


**Figure 2: Monthly precipitation, reference evapotranspiration from the CRUTS 4 database (NCAR, 2017) and derived water balance**
**for Salamat and Waza Logone.**


**Figure 3: Average Normalized Difference Vegetation Index (NDVI, MODIS 16 day interval and 250 m spatial resolution) measured**
**between 2003 and 2016 in the Salamat region and estimated monthly basal crop coefficient (Kcb, black points) for location S3.**






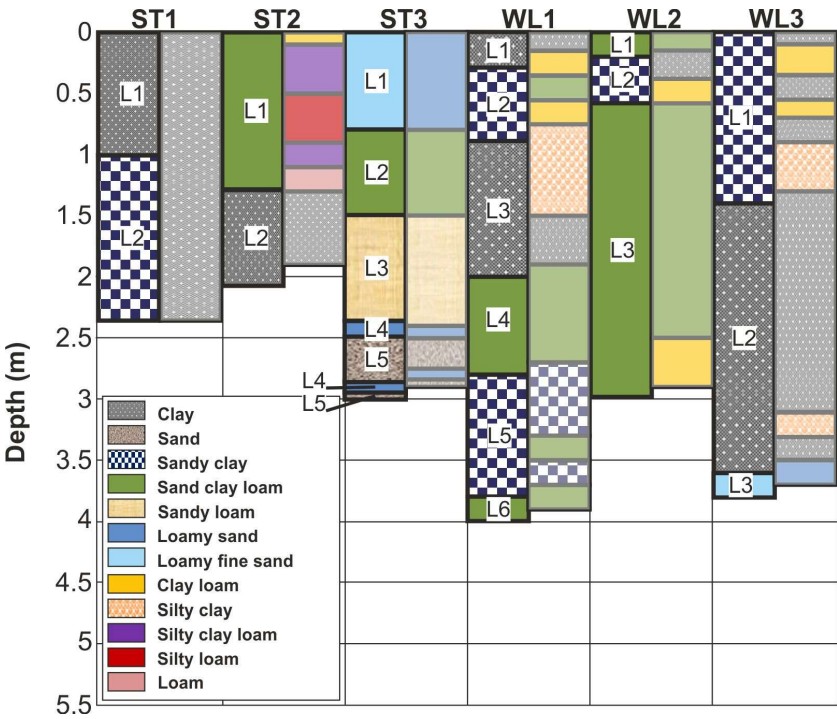


**Figure 4: Soil textures used in the model (left column) defined according to the grain size distribution analysis (right column) for**
**each of the six soil profiles.**

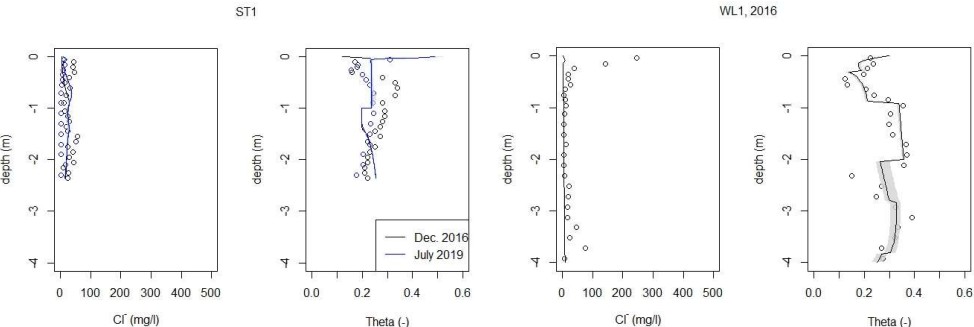






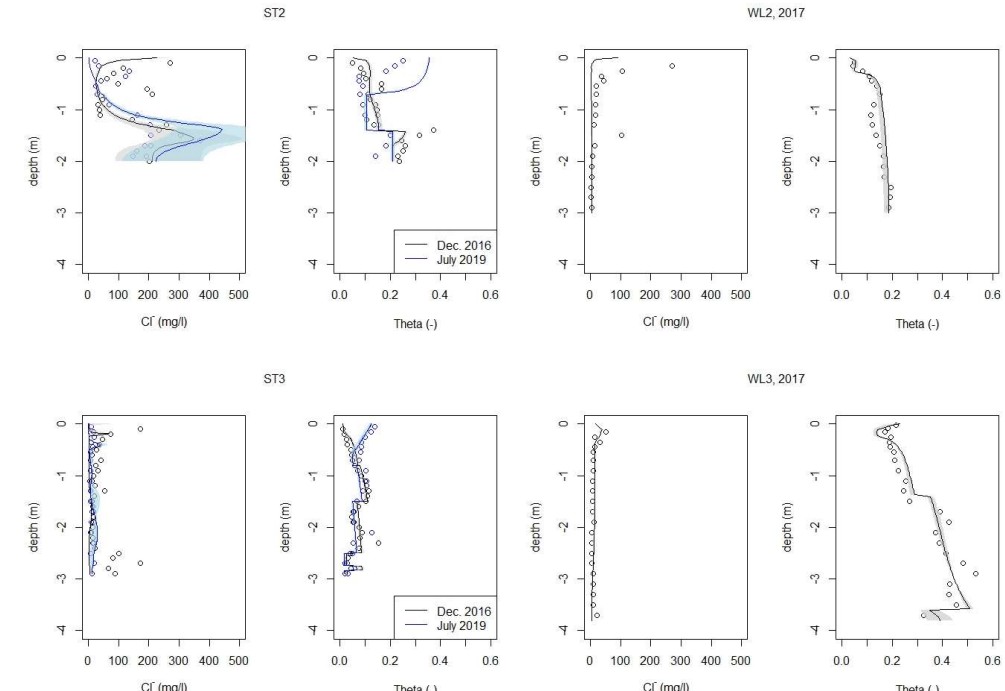



**Figure 5: Measured and simulated scenarios (for scenario definitions refer to Table 2) of chloride concentration and water content for all six soil profiles. Obs: Observations.**


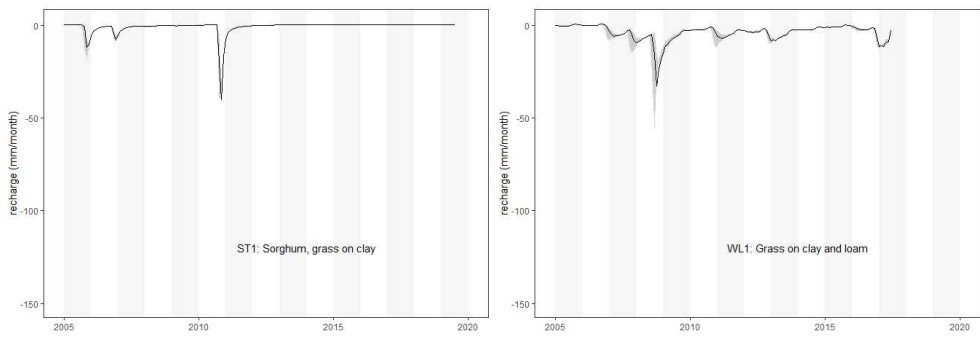





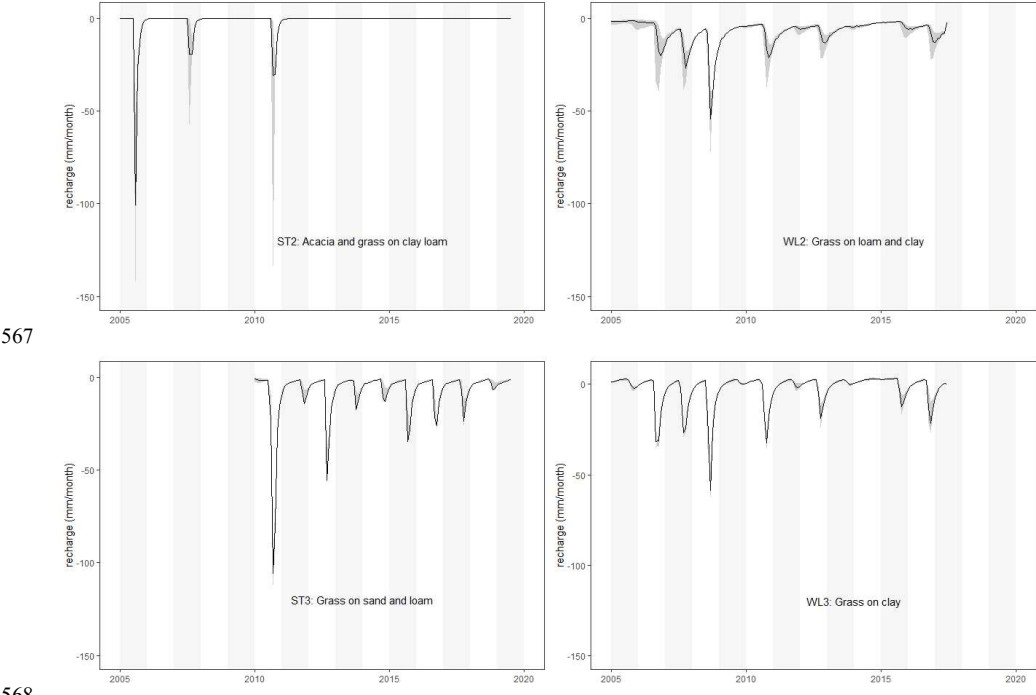


**Figure 6: Calculated groundwater recharge for all scenarios and sampling locations with indication of vegetation and soil texture.**



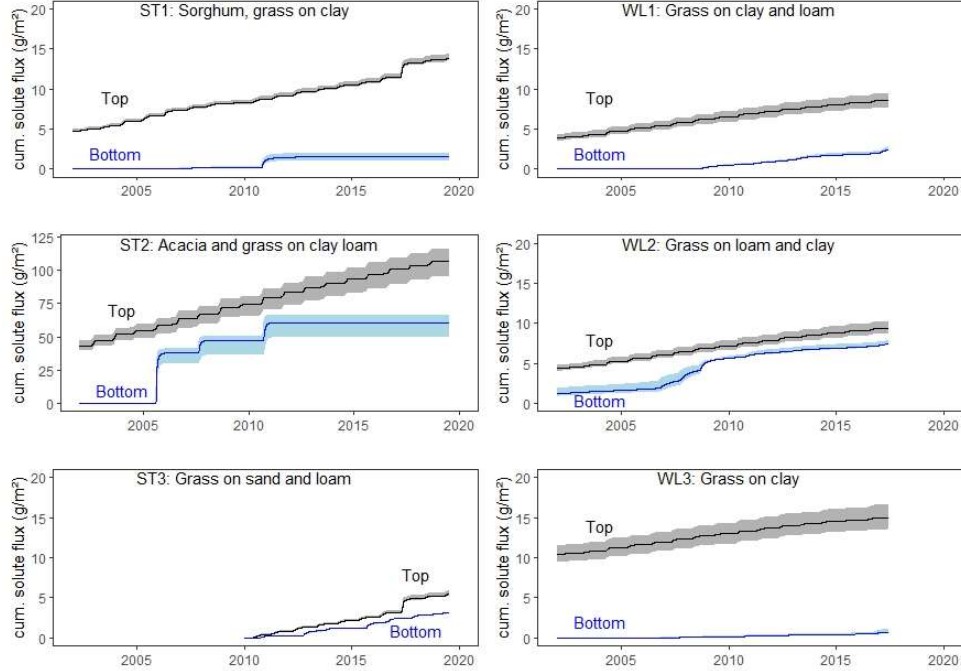


Fig. 7: Cumulative solute flux on the upper and lower boundary of the models. The shaded areas represents the standard deviation
of 100 randomly selected model runs. Note the different y-axis scales between sites.





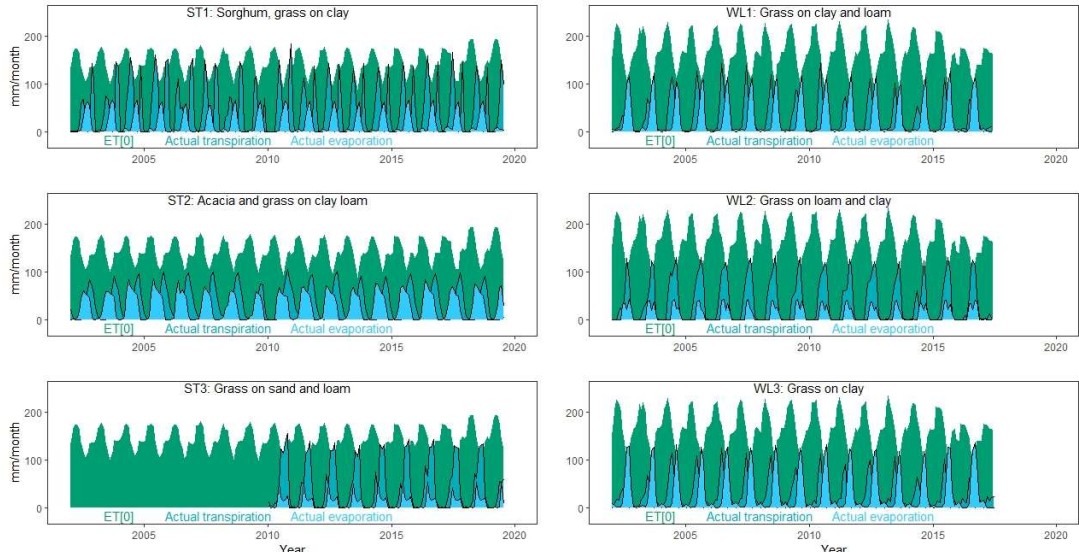

Fig. 8: Reference evapotranspiration from the CRUTS 4 database (NCAR 2017) as well as modelled average actual evaporation and transpiration of 100 randomly selected model runs.

Table 1: Names and geographic coordinates of the sampling locations with indication of average depth to groundwater.

| Name | Location | Date of Sampling | Drilling depth (m) | Longitude (°) | Latitude (°) | Elevation (m a. s.l.) | Depth to Groundwater (m) |
|---|---|---|---|---|---|---|---|
| ST1 | Gos Djarat | 07-12-2016 11-07-2019 | 2.35 5.0 | 19.89644 | 11.02582 | 418 | 21 |
| ST2 | Kach Kacha | 09-12-2016 16-07-2019 | 2.0 5.0 | 20.07473 | 10.81649 | 396 | 16-18 |
| ST3 | Gos Djarat | 11-12-2016 13-07-2019 | 2.2 5.0 | 19.91687 | 11.00629 | 418 | 21 |
| WL1 | Katoa | 01-06-2017 | 4.0 | 15.09235 | 10.82508 | 362 | 4 |
| WL2 | Loutou | 01-06-2017 | 3.0 | 15.37817 | 10.76805 | 325 | 11-12 |
| WL3 | Zina | 08-06-2017 | 3.8 | 14.97363 | 11.28858 | 304 | 3.6 |





**Table 2: Crop evapotranspiration scenario used with the individual soil profiles.**

| Scenario | Kcb | Ke | Root depth | Profile |
|---|---|---|---|---|
| Mean | average | average | average | All profiles |
| Min | minimum | minimum | average | All profiles |
| Min-RD | minimum | minimum | minimum | WL1 |
| Mix-1 | minimum | average | average | All profiles |
| Mix-2 | average | minimum | average | ST1, WL2, WL3 |
| Mix-3 | maximum | average | average | ST3 |
| Max | maximum | maximum | average | All profiles |


**Table 3: Parametrization of water retention and unsaturated hydraulic conductivity functions according the Mualem-van**
**Genuchten model after Bayesian model calibration.**

| Location | Texture | Depth (m) | $\theta_r$ (-) | $\theta_s$ (-) | $\alpha$ (m$^{-1}$) | n (-) | $K_s$ (md$^{-1}$) |
|---|---|---|---|---|---|---|---|
| ST1 | Clay | 0-1 | 0.001 | 0.61±0.01 | 2.13±0.27 | 1.164±0.008 | 0.09±0.14 |
| | Sandy clay | 1-2.35 | 0.04 | 0.43±0.03 | 2.63±0.37 | 1.150±0.011 | 0.43±0.39 |
| ST2 | Sandy clay loam | 0-1.4 | 0.04 | 0.38±0.02 | 1.18±0.08 | 1.36±0.047 | 0.03±0.16 |
| | Clay | 1.4 -2.1 | 0.07 | 0.48±0.08 | 2.66±0.36 | 1.203±0.052 | 0.11±0.28 |
| ST3 | Loamy fine sand | 0-0.8 | 0.01 | 0.45±0.08 | 3.69±0.08 | 2.332±0.196 | 2.96±5.72 |
| | Sandy clay loam | 0.8-1.5 | 0.043 | 0.38±0.07 | 2.81±0.43 | 2.210±0.172 | 2.44±4.19 |
| | Sandy loam | 1.5-2.4 | 0.02 | 0.43±0.08 | 3.44±0.51 | 2.469±0.330 | 1.66±2.84 |
| | Loamy sand | 2.4-2.5 | 0 | 0.35±0.06 | 3.77±0.53 | 1.980±0.265 | 2.03±3.11 |
| | Sand | 2.5-2.75 | 0 | 0.34±0.04 | 3.73±0.53 | 2.730±0.372 | 5.42±8.86 |
| | Loamy sand | 2.75-2.84 | 0 | 0.35±0.06 | 3.77±0.53 | 1.980±0.265 | 2.03±3.11 |
| | Sand | 2.84-2.9 | 0 | 0.34±0.04 | 3.73±0.53 | 2.730±0.372 | 5.42±8.86 |
| WL1 | Clay | 0-0.3 | 0.065 | 0.56±0.09 | 1.37±0.19 | 1.293±0.092 | 0.17±0.26 |
| | Sandy clay | 0.3-0.9 | 0.06 | 0.44±0.07 | 2.85±0.36 | 1.416±0.125 | 0.21±0.38 |
| | Clay | 0.9-2.0 | 0.103 | 0.42±0.03 | 1.55±0.21 | 1.187±0.065 | 0.19±0.42 |
| | Sandy clay loam | 2.0-2.8 | 0.075 | 0.49±0.07 | 2.34±0.33 | 1.598±0.227 | 0.13±0.28 |
| | Sandy clay | 2.8-3.8 | 0.081 | 0.43±0.06 | 2.60±0.35 | 1.266±0.134 | 0.09±0.19 |



| | Sandy clay loam | 3.8-4.0 | 0.071 | 0.40±0.05 | 2.69±0.37 | 1.291±0.137 | 0.12±0.24 |
|---|---|---|---|---|---|---|---|
| WL2 | Sandy clay loam | 0-0.2 | 0.03 | 0.41±0.07 | 3.22±0.45 | 1.502±0.151 | 0.30±0.57 |
| | Sandy clay | 0.2-0.6 | 0.01 | 0.37±0.06 | 2.56±0.39 | 1.422±0.081 | 0.09±0.19 |
| | Sandy clay loam | 0.6-3.0 | 0.01 | 0.37±0.03 | 1.39±0.19 | 1.566±0.06 | 0.10±0.10 |
| WL3 | Sandy clay | 0-1.4 | 0.09 | 0.49±0.09 | 1.27±0.15 | 1.470±0.111 | 0.22±0.14 |
| | Clay | 1.4-3.6 | 0.105 | 0.53±0.05 | 2.03±0.29 | 1.285±0.100 | 0.17±0.36 |
| | Loamy fine sand | 3.6-3.8 | 0.056 | 0.39±0.08 | 2.90±0.45 | 1.789±0.293 | 1.23±2.40 |


**Table 4: Average root mean square error (RMSE) and related standard deviation (SD) over all scenarios for water content (Theta)**
**and chloride concentration.**

| Location, Year | Theta (-) | | | Chloride concentration (mg/l) | | |
|---|---|---|---|---|---|---|
| | Average observation | Average simulation | Average RMSE | Average observation | Average simulation | Average RMSE |
| ST1, 2016/2019 | 0.25/0.22 | 0.23/0.23 | 0.06/0.04 | 30/6 | 18/22 | 19/19 |
| ST2, 2016/2019 | 0.17/0.14 | 0.16/0.15 | 0.06/0.04 | 162/106 | 132/229 | 82/116 |
| ST3, 2016/2019 | 0.06/0.08 | 0.07/0.06 | 0.02/0.02 | 42/10 | 6/13 | 58/10 |
| WL1, 2017 | 0.27 | 0.27 | 0.05 | 31 | 6 | 59 |
| WL2, 2017 | 0.13 | 0.15 | 0.02 | 40 | 3 | 117 |
| WL3, 2017 | 0.31 | 0.33 | 0.04 | 12 | 13 | 9 |


**Table 5: Calculated average annual recharge, fraction of recharge on average annual precipitation, standard deviations of recharge**
**across the time-period 2005-2019 and 2005 – 2016 for Salamat and Waza Logone respectively.**

| Location | Average annual recharge (mm) | Fraction of average annual precipitation (%) | Standard deviation of annual recharge (mm) |
|---|---|---|---|
| ST1 | 7 | 0.9 | 17 |
| ST2 | 9 | 1 | 29 |
| ST3 | 93 | 12 | 69 |
| WL1 | 28 | 4 | 32 |
| WL2 | 54 | 8 | 46 |
| WL3 | 6 | 1 | 48 |





**Table 6: Calculated average annual evaporation and transpiration and related standard deviations of 100 randomly accepted model**
**runs.**

| Location | Average annual evaporation (mm) | Standard deviation of evaporation (mm) | Average annual transpiration (mm) | Standard deviations of transpiration (mm) | Average actual evapotranspiration (mm) |
|---|---|---|---|---|---|
| ST1 | 210 | 9 | 553 | 11 | 763 |
| ST2 | 366 | 22 | 388 | 27 | 754 |
| ST3 | 137 | 12 | 552 | 11 | 689 |
| WL1 | 344 | 20 | 317 | 23 | 661 |
| WL2 | 146 | 14 | 477 | 28 | 623 |
| WL3 | 376 | 12 | 305 | 10 | 681 |
