# Peer review of "Modelling groundwater recharge, actual evaporation and"

_Hydrology and Earth System Sciences, 2021_

## Referee Comment (RC1)

**Revision of Manuscript hess-2021-390**

**Title: Modelling groundwater recharge, actual evaporation and**

**transpiration in semi-arid sites of the Lake Chad Basin: The role of**

**soil and vegetation on groundwater recharge**

The authors test an experimental approach for assessing groundwater recharge in Lake Chad Basin based on a small number of field data and available remote sensing products. This set of data consists of soil water content values and chloride concentrations at different soil depths over 6 experimental soil profiles. The low-cost data set is implemented in Hydrus-1D for simulating groundwater recharge

The evaluation of this manuscript is based on the following questions:

1) Is it a novel work based on a reliable scientific technique?
2) Is it clearly structured and well-written?
3) Are the experimental design and analysis of data adequate and appropriate to the investigation?

The authors should stress on the novelty of this paper. In my understanding, they try to provide an affordable low-cost approach in a data-poor region to assess groundwater recharge. Nevertheless, the description of Materials and Methods is poorly described, unclear and some parts are "gray". Precipitation and ET are given at monthly scale. The authors then declare that they set up Hydrus-1D at monthly scale (line 185). As far as I know, time units in Hydrus-1D are seconds, minutes, hours, days and years. The authors are invited to give more detailed information on the Hydrus-1D version. Did the authors use year fractions? In any case, running Hydrus-1D at monthly scale provides only a gross water balance simulation.

The manuscript is potentially interesting for the readers, however it needs substantial revisions before publication in light of the following comments:

1) I invite the authors to thoroughly revise Materials and Methods by adding a methodological sub-section in which they describe step-by-step the proposed approach. Maybe they can add a flowchart, a schematic overview to clarify all steps.
2) Session 2.5 is very unclear. I suggest to substantially revise this part. The authors declare that they have monthly P and ET from 1970 to 2019 (line 123). Then in line 192 they set up a burn-in period of 80 years to relax the impact of unknown initial conditions on model simulations. How can this be possible? In Fig. 6 I see simulations of groundwater recharge from 2005 up to 2019. I do not understand the impact of the 7 scenarios on model results.
3) In sub-Session 2.5.2 the authors use the bulk density in Rosetta in Hydrus-1D. How can you sample a known soil volume from the auger? Please clarify it in sub-session 2.3. It is recommended to add the Richards equation, the van Genuchten (1980) equation for soil

water retention function and hydraulic conductivity function by declaring all soil hydraulic parameters ($\theta$r, $\theta$s, n, $\alpha$ and Ks)

4) The description of scenarios and model calibration in section 2.5.5 and 2.5.6 is unclear at all. I don't know if I got the description. Since the authors do not know the values of Kcp, Ke and root depth, they organize 7 scenarios (Table 2). In each scenario they run inverse modeling to optimize the five unknown van Genuchten parameters ($\theta$r, $\theta$s, n, $\alpha$ and Ks) given in Table 3. The authors should list the parameters they optimize with the inverse modelling exercise in Session 2.5.6. The authors use Rosetta PTF to set initial parameter values. Then the calibration results are presented in Table 3 and Table 4 and Figure 5, 6 and 7.

5) Finally the authors try to poorly interpret ET dynamics in Section 3.4

Given the 5 points listed above, the manuscript seems a calibration exercise supported by scarce information. What is the novelty of this study?

Authors should mention in the conclusion where Hydrus-1D can be improved or any suggestions for future research in terms of models' performance improvement and application.

---

## Referee Comment (RC2)

[referee-annotated manuscript omitted]

---

## Referee Comment (RC3)

**Comments on manuscript HESS-2021-390**

**Modelling groundwater recharge, actual evaporation and transpiration in semi-arid sites of the Lake Chad Basin: The role of soil and vegetation on groundwater recharge**

**By Neukum et al.**

This paper presents an interesting approach for evaluating groundwater recharge in a remote semi-arid area. The study is based on soil water content and chloride content data, measured in selected study sites. Quantifying groundwater recharge is particularly crucial in the Sahelian part of West Africa, and considering the few number of studies in these areas, the data scarcity, and the difficult access to field work, the present study would deserve to be published. Nevertheless, I have different comments that should be addressed before publication.

**General comments**

In the introduction, a large part is given to the question of evapotranspiration partitioning, but this is not really discussed in the paper, and no mention of that is present in the conclusion. Authors should therefore either better discussed their results, or change the focus of the introduction (and title).

The paper from Tewolde et al 2019 presented an approach which is (at least partly) similar, involving most of the authors of the present manuscript, and some of the soil data seem to derive from the same database. Therefore, the differences between the two papers, their complementarity, and the added values of the present study, involving a modeling approach, should be described in the introduction.

The authors should also define what is considered under the term « recharge », since the soil water infiltration may not reach the water table. In addition, the depth of transpiration uptake can overcome the maximum depth of the soil profiles and affect the effective groundwater recharge.

The evaluation of recharge rates obtained from this study should be discussed and compared with previous evaluations in these areas, or under similar environment (e.g. Tewolde 2019, but nor only). A discussion of the spatial significance of the results is lacking. In such arid environments, recharge is mainly focused, and strongly controlled by surface hydrology (e.g. flooding episodes). Results regarding the variations of recharge rates in relation to surface hydrological conditions, would greatly help to go further in up-scaling of recharge rates.

Regarding the time variations of recharge fluxes (interannual *versus* seasonal), a strong impact of the extension of flooded areas is expected. The study sites are located in flooding plains, which involve a "binary" behavior of water infiltration, between periods of saturated soil surface, and non-saturated periods. During submersion phases, a constant maximum flux of water infiltration could be estimated? This does not appear in Figure 5.

**Description of data and method**

This section doesn't allow to fully understand the data and method and needs to be rewritten.

Details are needed on the choice of study sites, their situation regarding surface hydrology and flooding occurrences.

A dedicated "geochemical data" section is needed. Regarding the chloride concentrations of rainfall water, the authors should describe the rainfall collectors (always open or only during rainfall events? It is important for dry deposition), and the water analysis and precision, a crucial information for these very low concentrations. I would suggest including a table with the concentration data and corresponding rainfall amounts. It is suggested that the rainfall collection was not exhaustive (42 samples between 2014 and 2020). Therefore, an evaluation of the "missing" chloride input should be discussed. What was the proportion of non-sampled rainfall?

Two sampling locations of rainfall water are mentioned, how did they compare? The atmospheric chloride inputs could also be compared with previous regional evaluation in tropical Africa (e.g. Laouali et al 2012).

Regarding the chloride concentrations of soil water, detailed information is lacking on the different soil profiles collected: sampling dates and locations, the depth of the saturated zone, the distance between soil profiles of a same site, etc... Are the chloride data different from those published by Tewolde et al 2019? Which data were previously published, and which data are new? Why was the profile ST4 from Tewolde et al 2019 discarded? It should be explained in the data section. Here also a table would be useful.

The description of the methods needs to be clarified.
Regarding the chloride mass balance, describe the assumptions, the equations, etc... Storage of chloride in the vadose zone is important and should be discussed regarding steady-state *vs* transient state assumptions.

Regarding the FAO-dual Kc concept, it is not clear how it was integrated in the 1D modelling approach.

Explain the "scenarios". I have the feeling that it is a kind of sensitivity analysis, please explain.

**Detailed comments**

l. 20: What is a "potential recharge rate" ?

l.129-130 : what explains the higher value of potential evapotranspiration in the Salamat site, located at the same latitude as the Logone site? Local environment? Is the difference greater than uncertainty?

l. 171: I didn't find table S1: From my uploaded file, the supplementary document only included a truncated Table, without titles.

l.203 : explain which data were used to define the upper boundary conditions. Did they include the full saturation of the top soil?

l.227-228: unclear. What is a "Cauchy" boundary condition?

l. 229-230: a first reference is made here to the sampling of ponding water. It should be described in the data section

l.231: I didn't understand the definition of "ET scenarios". Explain which are the corresponding assumptions and objectives.

Figure 5 : The caption is not complete (Obs ? profile dates ?)

Figure 6: indicate the sampling periods. Also, it would be interesting to add rainfall data together with recharge rates, if possible.

Figure 7: the contrast between the top and bottom solute fluxes is huge. It implies a strongly intransient behavior, and accumulation of chloride. Please comment on that, regarding the long-term behavior and the resulting soil salinization. Also, what could be expected from soil leaching during flood events?

Figure 8: not sure if this figure is useful

**References cited**

Laouali, D., Galy-Lacaux, C., Diop, B., Delon, C., Orange, D., Lacaux, J.P., Akpo, A., Lavenu, F., Gardrat, E., Castera, P., 2012. Long term monitoring of the chemical composition of precipitation and wet deposition fluxes over three Sahelian savannas. Atmos. Environ. 50, 314–327. https://doi.org/10.1016/j.atmosenv.2011.12.004

Tewolde, D.O., Koeniger, P., Beyer, M., Neukum, C., Groeschke, M., Ronelngar, M., Rieckh, H., Vassolo, S., 2019. Soil water balance in the Lake Chad Basin using stable water isotopes and chloride of soil profiles. Isot. Environ. Health Stud. 55, 459–477. https://doi.org/10.1080/10256016.2019.1647194

---

## Author Comment (AC2)

[revised manuscript text omitted]

- 545

---

## Author Comment (AC3)

**Comments on manuscript HESS-2021-390**

**Modelling groundwater recharge, actual evaporation and transpiration in semi-arid sites of the Lake Chad Basin: The role of soil and vegetation on groundwater recharge**

**By Neukum et al.**

This paper presents an interesting approach for evaluating groundwater recharge in a remote semi-arid area. The study is based on soil water content and chloride content data, measured in selected study sites. Quantifying groundwater recharge is particularly crucial in the Sahelian part of West Africa, and considering the few number of studies in these areas, the data scarcity, and the difficult access to field work, the present study would deserve to be published. Nevertheless, I have different comments that should be addressed before publication.

**General comments**

In the introduction, a large part is given to the question of evapotranspiration partitioning, but this is not really discussed in the paper, and no mention of that is present in the conclusion. Authors should therefore either better discussed their results, or change the focus of the introduction (and title).

*Sub-section 3.1 is dedicated to partitioning of evapotranspiration. A sentence discussing the issue of partitioning has been added to the Discussion.*

The paper from Tewolde et al 2019 presented an approach which is (at least partly) similar, involving most of the authors of the present manuscript, and some of the soil data seem to derive from the same database. Therefore, the differences between the two papers, their complementarity, and the added values of the present study, involving a modeling approach, should be described in the introduction.

*The last sentence has been completed to explain that part of the data from Tewolde et al. (2019) has been used in this publication to try to produce time series of recharge for the period 2003-2016.*

The authors should also define what is considered under the term « recharge », since the soil water infiltration may not reach the water table. In addition, the depth of transpiration uptake can overcome the maximum depth of the soil profiles and affect the effective groundwater recharge.

*We consider recharge the water that percolates below the root zone. We have set the root depths for each vegetation type (Table S1) according to data published be Feddes et al. (1978). Depth to groundwater is presented in Table 1. Vegetation roots were largely located within the unsaturated zone in all none of the studied profiles.*

The evaluation of recharge rates obtained from this study should be discussed and compared with previous evaluations in these areas, or under similar environment (e.g. Tewolde 2019, but nor only). A discussion of the spatial significance of the results is lacking. In such arid environments, recharge is mainly focused, and strongly controlled by surface hydrology (e.g. flooding episodes). Results regarding the variations of recharge rates in relation to surface hydrological conditions would greatly help to go further in up-scaling of recharge rates.

*Comparison with published data for similar environments has been added to the discussion. According to our experience, soil texture plays a larger role than surface hydrology when it comes to recharge. We have modelled five locations that are annually flooded. However, the largest annual mean recharge was obtained for site ST3, which is never flooded (compare Table 5).*

Regarding the time variations of recharge fluxes (interannual versus seasonal), a strong impact of the extension of flooded areas is expected. The study sites are located in flooding plains, which involve a "binary" behavior of water infiltration, between periods of saturated soil surface, and non-saturated periods. During submersion phases, a constant maximum flux of water infiltration could be estimated? This does not appear in Figure 5.

*Our experience shows that soils in flooded areas of the LCB are characterized by very fine material (loam to clay). Floodwater is retained by the soils to be later either evaporated or used by vegetation to grow. In the case of coarse soils like in ST3, which is located outside the wetlands, precipitation water is not retained by the soil and percolates almost directly producing recharge. These effects are clearly seen in Figure 6.*

**Description of data and method**

This section doesn't allow to fully understand the data and method and needs to be rewritten.
*Done*

Details are needed on the choice of study sites, their situation regarding surface hydrology and flooding occurrences.
*Done*

A dedicated "geochemical data" section is needed. Regarding the chloride concentrations of rainfall water, the authors should describe the rainfall collectors (always open or only during rainfall events? It is important for dry deposition), and the water analysis and precision, a crucial information for these very low concentrations. I would suggest including a table with the concentration data and corresponding rainfall amounts. It is suggested that the rainfall collection was not exhaustive (42 samples between 2014 and 2020). Therefore, an evaluation of the "missing" chloride input should be discussed. What was the proportion of non-sampled rainfall?
*Included in the text under sub-section 2.2*

Two sampling locations of rainfall water are mentioned, how did they compare? The atmospheric chloride inputs could also be compared with previous regional evaluation in tropical Africa (e.g. Laouali et al 2012).
*Precipitation chloride used in the modelling activities belong to only one location: N'Djamena. The text has been corrected accordingly (Sub-section 2.2, lines 134-141). Atmospheric chloride was compared with values from Laouali et al. (2012) and lay in the same range. Accordingly written in the text.*

Regarding the chloride concentrations of soil water, detailed information is lacking on the different soil profiles collected: sampling dates and locations, the depth of the saturated zone, the distance between soil profiles of a same site, etc... Are the chloride data different from those published by Tewolde et al 2019? Which data were previously published, and which data are new? Why was the profile ST4 from Tewolde et al 2019 discarded? It should be explained in the data section. Here also a table would be useful.
*Information on sampling dates and locations, the depth of the saturated zone, the distance between soil profiles of a same site, etc. are listed in Table 1. Chloride data measured in 2016 and 2017 correspond to those published by Tewolde et al. (2019). Measurements from 2019 are new data. It has been accordingly explained in the text.*
*ST4 was discarded, because it is located far from any floodplain and has characteristics similar to ST3. Including ST4 would not provide for additional information. Explained in sub-section 2.1, lines 115-118.*

The description of the methods needs to be clarified. Regarding the chloride mass balance, describe the assumptions, the equations, etc… Storage of chloride in the vadose zone is important and should be discussed regarding steady-state vs transient state assumptions.

*The sub-section describing methods has been rewritten to provide for more information. Assumptions and equations for chloride mass balance have been added (sub-sections 3.2.1 and 3.2.2).*

*Chemical memory effects are subject to the dynamics of the water and chloride balance. Therefore, accurate estimations are only possible with transient assumptions. Steady-state assumptions are unsuitable (sub-section 4.3, lines 364-365..*

Regarding the FAO-dual Kc concept, it is not clear how it was integrated in the 1D modelling approach.
*The FAO-dual Kc concept was used to be able to separate E from T in evapotranspiration. These parameters are used as input in the 1-D modelling (compare section 3).*

Explain the "scenarios". I have the feeling that it is a kind of sensitivity analysis, please explain.
*Scenarios are explained in sub-section 3.2.5*

**Detailed comments**

l. 20: What is a "potential recharge rate" ? *we meant the recharge that could take place. The word potential has been deleted.*

l.129-130 : what explains the higher value of potential evapotranspiration in the Salamat site, located at the same latitude as the Logone site? Local environment? Is the difference greater than uncertainty? *It is due to the coarser soil texture. It is explained in the text*

l. 171: I didn't find table S1: From my uploaded file, the supplementary document only included a truncated Table, without titles. *We are sorry. The full table was provided. Something must have gone wrong*

l.203 : explain which data were used to define the upper boundary conditions. Did they include the full saturation of the top soil? *Through the upper boundary, atmospheric water and solute are input into the coil column. The upper boundary conditions was defined as variable atmospheric condition with surface runoff as well as a time-dependent concentration*

l.227-228: unclear. What is a "Cauchy" boundary condition? *It has been clarified in the text (Line 283)*

l. 229-230: a first reference is made here to the sampling of ponding water. It should be described in the data section. *Done*

l.231: I didn't understand the definition of "ET scenarios". Explain which are the corresponding assumptions and objectives. *They are described in detail in sub-section 3.2.5*

Figure 5 : The caption is not complete (Obs ? profile dates ?) *Caption has been corrected (Obs was a remain from an older version. We apology for this)*

Figure 6: indicate the sampling periods. Also, it would be interesting to add rainfall data together with recharge rates, if possible. *We prefer to publish figure 6 as it is to provide for a better visibility of results.*

Figure 7: the contrast between the top and bottom solute fluxes is huge. It implies a strongly intransient behavior, and accumulation of chloride. Please comment on that, regarding the long-term behavior and the resulting soil salinization. Also, what could be expected from soil leaching during flood events? *Described in sub-section 4.3 as well as in section 5 lines 399-405*

Figure 8: not sure if this figure is useful. *We think that figure 8 is necessary. It visualizes the fact that the actual evapotranspiration is lower than the reference one. Actual evapotranspiration rate depends on water availability in the upper soil zone. Evaporation is very high in clay and clay-loam soils with higher water storativity and the upper part of the soils dries completely every year during the dry season. Only during or short after the rainy season, the real evapotranspiration becomes comparable to the reference one.*